# Latent Reasoning in TRMs is Secretly a Policy Improvement Operator

**Arip Asadulaev** [1]  **Rayan Banerjee** [1]  **Fakhri Karray** [1]  **Martin Takac** [1]

## Abstract

Recently, small models with latent recursion have obtained promising results on complex reasoning tasks. These results are typically explained by the theory that such recursion increases a network's depth, allowing it to compactly emulate the capacity of larger models. However, the performance of recursively added layers remains behind the capabilities of one-pass models with the same feed-forward depth. This means that in the looped version, not every recursive step effectively contributes to depth. This raises the question: when and why does latent reasoning improve performance, and when does it result in *dead compute?* In our work, we demonstrate that latent recursive reasoning provides answer to this question. We show that latent recursive reasoning can be formalized as a policy improvement algorithm. Building on these insights, we propose to use a training schemes from reinforcement learning and diffusion methods for latent reasoning models. Using the Tiny Recursive Model as our testbed, we show that with our modifications we can avoid dead compute steps and reduce the total number of forward passes by 18× while maintaining performance. Broadly speaking, we show how a policy improvement perspective on recursive steps can explain model behavior and provide insights for further improvements.

## 1. Introduction

The goal of reasoning models is to start from a set of specific examples or observations and to infer a general rule or pattern. Such models systematically manipulate and connect pieces of information to infer new conclusions and solve problems not explicitly stated in the training data.

Recently, the use of latent reasoning and iterative refinement loops has become a major driver of progress in deep learning reasoning. Looped transformers (Giannou et al., 2023; Yang et al., 2023) repeatedly apply the same transformer block with input injection at each step and achieve better performance than a standard forward pass transformer in reasoning and meta learning tasks, while using a 10x lower number of parameters (Yang et al., 2023).

Recently, models like Hierarchical Reasoning Models (HRM) (Wang et al., 2025) and Tiny Recursive Models (Jolicoeur-Martineau, 2025), were built on the similar idea of reusing model output repeatedly in both the input and latent spaces. These models have demonstrated impressive performance and even outperformed multi-billion parameter LLM models on complicated `ARC-AGI` (Chollet, 2019).

In our paper we ask a question when an why is a simulated effective layer really effective? And after answering this, we want to know a better path on how such a model can be improved. This latent reasoning does some algorithm using the steps in latent space, but what kind of algorithm is actually running in this latent space. In our paper, we are not seeking for some interoperability of the latent representation, our goal is to formally bridge the algorithm of latent space reasoning to the well-known frameworks.

For this, we demonstrate how latent reasoning facilitates the policy improvement algorithm that aims to improve an *advantage margin* between the one-step output and the recurrence one (Frans et al., 2025). based on our findings, we propose an improved algorithm that makes the reasoning process more task-oriented.

TRMs are difficult to train efficiently: supervision is primarily applied to the final output, leaving early refinement steps with weak credit assignment and making optimization brittle in the small-data regime. We propose *Deep Improvement Supervision* (DIS), a training method that supplies step-wise intermediate targets during recursion. In our main implementation, these targets are generated by a monotonic discrete corruption schedule of the ground-truth output, turning each recursion step into a supervised sub-goal.

Our method achieves competitive performance on complex reasoning benchmarks, including `ARC-AGI 1`, `ARC-AGI 2`, using a much simpler architecture than TRM. We avoid training the halting step module, use 3x fewer

---

[1]MBZUAI. Correspondence to: Arip Asadulaev <arip.asadulaev@mbzuai.ac.ae>.

supervision steps and 8x fewer latent reasoning steps. As a highlight of our method, our model achieves a 24% accuracy on `ARC-AGI-1` with 0.8 million parameters, outperform most of the existing open source LLM models without any external knowledge.

## 2. Background

### 2.1. Hierarchical Reasoning Models

A looped (Giannou et al., 2023) and universal (Dehghani et al., 2018) transformer repeatedly applies the same transformer block, with input injection each time, and is trained to make its intermediate loop output correct. This model was applied for various tasks, showing an improvement over the single step models (Yang et al., 2023).

Based on this idea, Hierarchical Reasoning Models (HRMs) (Wang et al., 2025) are supervised sequence-to-sequence models that perform *recursive refinement* of a prediction by interleaving two small recurrent networks that operate at different update frequencies. Let $\tilde{\mathbf{x}} \in \mathcal{V}^L$ denote an input sequence of length $L$ in a vocabulary $\mathcal{V}$, and let $\mathbf{y} \in \mathcal{V}^L$ be the desired output. HRM uses an input embedding $f_I$, two recurrent reasoning modules, a *low-level* module $f_L$ and a *high-level* module $f_H$ and an output head $f_O$.

After embedding $\mathbf{x} = f_I(\tilde{\mathbf{x}}) \in \mathbb{R}^{L \times D}$, HRM carries two latent states $\mathbf{z}_L, \mathbf{z}_H \in \mathbb{R}^{L \times D}$ through the supervision steps. Within a forward pass, it performs $n$ updates of $f_L^\phi$ for every update of $f_H^\psi$ and repeats this $T$ time before decoding with $f_O$. A typical schedule used in previous work is $n = 2$ and $T = 2$. The final prediction is $\hat{\mathbf{y}} = \arg\max f_O(\mathbf{z}_H)$. During a forward pass, HRM evaluates the following updates:

$$\mathbf{z}_L^{t+1} \leftarrow f_L^\phi((\mathbf{z}_L^t + \mathbf{x}) + \mathbf{z}_H^t), \quad \text{(repeated } t \text{ times)} \quad (1)$$
$$\mathbf{z}_H^{t+1} \leftarrow f_H^\psi(\mathbf{z}_L^{t+1} + \mathbf{z}_H^t),$$
$$\hat{\mathbf{y}} = \arg\max f_O(\mathbf{z}_H).$$

Most evaluations in the early part of the schedule are executed without gradient tracking, while the final evaluations are backpropagated through. This design aims to amortize compute while allowing the model to refine internal states before a gradient-bearing step.

**Deep supervision.** To emulate very deep computation without prohibitive memory, HRM reuses $(\mathbf{z}_L, \mathbf{z}_H)$ across $N_{\text{sup}}$ supervision steps up to 16, detaching the states between steps. This *deep supervision* improves the answer iteratively and yields hundreds of effective layers, while avoiding full backpropagation over time.

**Adaptive Computational Time.** Training-time efficiency is improved by a learned halting mechanism (ACT). A small head predicts whether to stop iterating on the current example or continue; the published implementation trains this with a halting loss and an additional continue loss that

requires an *additional* forward pass, effectively doubling forward compute per optimization step (Graves, 2016). The test-time evaluation runs a fixed maximum number of supervision steps to maximize accuracy.

### 2.2. Tiny Recursive Models

Tiny Recursive Models (Jolicoeur-Martineau, 2025) retain the core idea of iterative refinement but collapse HRM's complexity into a *single* tiny network and a simpler recursion scheme. In the TRM setup, $\mathbf{z}_H$ is the state that the model reads out to produce the answer (the output head is applied to $\mathbf{z}_H$: $\hat{\mathbf{y}} = \arg\max f_O(\mathbf{z}_H)$). $\mathbf{z}_L$ is a *working memory* state that is updated using the input $\mathbf{x}$ and the current answer, and is then used to update $\mathbf{z}_H$. Because the loss is applied on the prediction from $\mathbf{z}_H$, optimization pressure makes $\mathbf{z}_H$ look like (encode) the current solution.

On the other hand, $\mathbf{z}_L$ is only indirectly supervised through its effect on $\mathbf{z}_H$, so it is free to be an internal reasoning representation rather than a decodable solution. Within same $f^\phi$ network for $L$ and $H$ module, TRM repeats each recursion equal to HRM Eq. 1.

The prediction is made from $\mathbf{z}_H$ through the output head and trained with cross-entropy. This asymmetry (only $\mathbf{z}_H$ sees final $\mathbf{z}_L^{n+1}$; and only $\mathbf{z}_H$ is decoded and penalized) naturally pushes $\mathbf{z}_H$ towards the space of valid answers, while $\mathbf{z}_L$ becomes the latent *reasoning scratchpad* that helps improve the next $\mathbf{z}_H$.

The TRM paper explicitly reframes this: $\mathbf{z}_H$ *is simply the current (embedded) solution...* $\mathbf{z}_L$ *is a latent feature that does not directly correspond to a solution but can be transformed into one by* $f_H$. It was shown on the Sudoku example that when you `reverse-embed + argmax`, the tokenized $\mathbf{z}_H$ looks like the solved grid, while tokenized $\mathbf{z}_L$ is not realted to the solution.

From the notation perspective, we want to note that TRM renames $\mathbf{z}_H$ to $\mathbf{y}$ (the running answer) and $\mathbf{z}_L$ to $\mathbf{z}$ (latent reasoning). The loop becomes: update $\mathbf{z}$ using $(\mathbf{x}, \mathbf{y}, \mathbf{z})$; then update $\mathbf{y}$ using $(\mathbf{y}, \mathbf{z})$. Carrying both across deep-supervision steps lets the model iterate: $\mathbf{z}$ remembers how it got to the current guess (like a chain-of-thought), and $\mathbf{y}$ stores the current guess itself. TRM trains a *single* halting probability via binary cross-entropy against correctness.

Let $\mathcal{L}_{\text{task}} = \text{CE}(f_O(\mathbf{y}), \mathbf{y}_{\text{true}})$ be the prediction cross-entropy loss and $\mathcal{L}_{\text{halt}} = \text{BCE}(q(\mathbf{y}), \hat{\mathbf{y}} = \mathbf{y}_{\text{true}})$ the halting loss with $q(\cdot)$ a scalar head. An optimization step iterates up to $N_{\text{sup}}$ supervision steps, performing $T - 1$ no-grad recursion cycles, then one with gradients, detaching $(\mathbf{y}, \mathbf{z})$ between supervision steps. Early stopping within a mini-batch is permitted by using the halting signal.

## 2.3. Policy Improvement Algorithm

We consider a Markov decision process with state space $\mathcal{S}$, action space $\mathcal{A}$, and a stochastic policy $\pi(a \mid s)$. Given a reference policy $\hat{\pi}$, we define the discounted state-value and action-value functions as $V^{\hat{\pi}}(s)$ and $Q^{\hat{\pi}}(s,a)$, respectively. The advantage function is defined as

$$A^{\hat{\pi}}(s,a) = Q^{\hat{\pi}}(s,a) - V^{\hat{\pi}}(s).$$

A policy improvement operator maps a reference policy $\hat{\pi}$ to a new policy $\pi$ so that the expected return does not decrease, i.e. $J(\pi) \geq J(\hat{\pi})$. A sufficient condition for policy improvement is that the expected advantage under the discounted state–action occupancy measure induced by $\pi$ is non-negative.In practice, policies are optimized using samples drawn from the reference policy $\hat{\pi}$. This leads to the approximate objective

$$\tilde{J}(\pi) = \mathbb{E}_{s \sim p_{\hat{\pi}}(s)} \left[ \mathbb{E}_{a \sim \pi(a|s)} \left[ A^{\hat{\pi}}(s,a) \right] \right], \quad (2)$$

where $p_\pi(s)$ denotes the *discounted* visitation distribution under $\pi$ (Sutton et al., 1998). To control deviations from the policy, it is common to introduce a KL regularization:

$$J(\pi) = \mathbb{E}_{\tau \sim p(\tau|\pi)} \left[ \sum_t \gamma^t r(s_t, a_t) \right]$$
$$- \beta \, \mathbb{E}_{s \sim p_\pi(s)} \left[ D_{\mathrm{KL}}(\pi(\cdot \mid s) \,\|\, \hat{\pi}(\cdot \mid s)) \right], \quad (3)$$

where $\beta > 0$ controls the tradeoff between reward maximization and adherence to $\hat{\pi}$. The solution to the KL-regularized policy improvement problem in Equation (3) takes the form of a product policy:

$$\pi(a \mid s) \propto \hat{\pi}(a \mid s) \exp\left( \tfrac{1}{\beta} A^{\hat{\pi}}(s,a) \right), \quad (4)$$

which corresponds to exponentiated advantage reweighting. This shows that *policy improvement can be achieved by multiplicatively reweighting a reference policy using any monotonic function of advantage*, providing a unifying view of trust-region methods, advantage-weighted regression, and KL-regularized reinforcement learning objectives.

Recently, it was shown that the policy improvement framework does not necessarily rely on explicitly learning a value function. Recent work formalizes a tight connection between *classifier-free guidance* (CFG) in diffusion/flow models and *policy improvement* in reinforcement learning (Frans et al., 2025). This work provides a framework for the analysis of diffusion models as RL methods. For a state $\mathbf{s}$ and an action/output $\mathbf{a}$, the classifier-free guidance RL (CFGRL) method parameterizes a target policy as:

$$\pi(a \mid \mathbf{s}) \propto \hat{\pi}(a \mid \mathbf{s}) \, f\big( A_{\hat{\pi}}(\mathbf{s}, \mathbf{a}) \big), \quad (5)$$

where $f : \mathbb{R} \to \mathbb{R}_{\geq 0}$ is a nonnegative monotonically increasing function of the advantage $A^\pi(\mathbf{s}, \mathbf{a}) = Q^\pi(\mathbf{s}, \mathbf{a}) - V^\pi(\mathbf{s})$ (Sutton et al., 1998), and $\hat{\pi}$ is a reference policy.

## 3. Formalization of Latent Reasoning as Policy Improvement

Both TRMs and HRMs are often motivated by *effective depth*: reusing a small parameter-shared network is said to emulate very deep computation with few parameters. In HRMs this story is tied to Deep Equilibrium (DEQ) models, where one assumes convergence to a fixed point and invokes the Implicit Function Theorem to justify differentiating through only the last iterate (Bai et al., 2019). However, follow-up analyses indicate that the fixed-point assumption is typically violated in practice: latent residuals remain non-negligible under truncation, and a one-step DEQ approximation does not fully explain the empirical gains (Jolicoeur-Martineau, 2025).

This paper adopts a different organizing principle. Rather than asking whether recursion converges, we ask: *what does one latent-reasoning step do to the model's output distribution?* Our central claim is that a single TRM step implements a *policy improvement* update: it takes a *reference* policy (the model's current guess) and transforms it into an *improved* policy using an internally computed signal.

**Notation.** At recursion step $t$ we write the TRM state as

$$s_t \;=\; (x, \mathbf{z}_L^t, \mathbf{z}_H^t),$$

where $\mathbf{x}$ is the embedded input, $\mathbf{z}_L^t$ is a latent *scratchpad*, and $\mathbf{z}_H^t$ is the embedding of the current solution. A parameter-shared backbone is used in two roles:

$$\mathbf{z}_L^{t+1} = f_\phi^{(L)}(\mathbf{x}, \mathbf{z}_H^t, \mathbf{z}_L^t), \qquad \mathbf{z}_H^{t+1} = f_\phi^{(H)}(\mathbf{z}_H^t, \mathbf{z}_L^{t+1}),$$
$$(6)$$

and an output head $f_O$ maps $\mathbf{z}_H$ to logits over the next token/action set $\mathcal{A}$. Only $\mathbf{z}_H$ is directly decoded and trained with cross-entropy.

### 3.1. Two policies per recursion answer

Each recursion step produces two output distributions at essentially no extra cost: one *before* the latent update and one *after* it.

**Reference policy (no-reasoning).** Forward decoding:

$$\ell_u^t \;:=\; f_O(\mathbf{z}_H^t), \qquad \hat{\pi}_t(a \mid s_t) \;:=\; \mathrm{softmax}(\ell_u^t). \quad (7)$$

**Improved policy (post-reasoning).** Decode after one latent-reasoning update:

$$\ell_c^t \;:=\; f_O(\mathbf{z}_H^{t+1}), \qquad \pi_t^+(a \mid s_t) \;:=\; \mathrm{softmax}(\ell_c^t). \quad (8)$$

Intuitively, $\hat{\pi}_t$ is the model's *first guess* at step $t$, while $\pi_t^+$ is the refined distribution after one reasoning step. We now introduce the same object used in policy-improvement analyses: an *optimality* (or *improvement*) variable. Let

$o \in \{0,1\}$ indicate whether an action $a$ is *good*[1] for state $s$. A standard KL-regularized improvement step produces a new policy by reweighting a reference policy $\hat{\pi}$ using an *optimality likelihood* $p(o = 1 \mid s, a)$:

$$\pi_w(a \mid s) \propto \hat{\pi}(a \mid s)\, p(o = 1 \mid s, a)^{w}, \qquad (9)$$

where $w \geq 0$ controls the step size: larger $w$ places more weight on actions deemed optimal. Equation (9) is exactly the multiplicative form of an exponentiated-advantage update; explicitly, if we define the *improvement score*

$$A(s, a) := \log p(o = 1 \mid s, a), \qquad (10)$$

then (9) is equivalent to

$$\pi_w(a \mid s) = \frac{\hat{\pi}(a \mid s)\, \exp\big(w\, A(s, a)\big)}{\mathbb{E}_{a' \sim \hat{\pi}(\cdot \mid s)}\big[\exp\big(w\, A(s, a')\big)\big]}. \qquad (11)$$

This view matches a general policy-improvement theorem for *product policies*. If we form a new policy by multiplying a reference policy by a non-negative, monotonically increasing function of advantage, the result is guaranteed to improve expected return. Let $f : \mathbb{R} \rightarrow \mathbb{R}_{\geq 0}$ be non-negative and monotonically increasing. If $A^{\hat{\pi}}(s, a)$ denotes the advantage under $\hat{\pi}$, then the product policy

$$\pi(a \mid s) \propto \hat{\pi}(a \mid s)\, f\big(A^{\hat{\pi}}(s, a)\big) \qquad (12)$$

is an improvement over $\hat{\pi}$. Please, see *Remark 1 and Theorem 1* (Frans et al., 2025). We can say that (11) is a special case of (12) with $f(u) = \exp(wu)$.

The only missing ingredient is: *what is $p(o = 1 \mid s, a)$ for latent reasoning?* In our setting, we will recover an *implicit* improvement factor $p(o = 1 \mid s, a)$ from the TRM's own post-reasoning policy, yielding the same product-policy form, but without explicitly learning a value function.

### 3.2. Improvement from a conditional policy

Here is the key observation: TRM already gives us a natural candidate for the *optimality-conditioned* policy. Interpret the post-reasoning distribution as

$$\pi_t^+(a \mid s_t) \approx p(a \mid s_t, o = 1), \qquad (13)$$

This relationship is evident in Eq. 6, where the updated policy $\pi_t^+$ depends on $\mathbf{z}_H^{t+1}$ which is updated via $\mathbf{z}_H^{t+1} = f_\phi^{(H)}(\mathbf{z}_H^t, \mathbf{z}_L^{t+1})$. Crucially, $\mathbf{z}_L^{t+1} = f_\phi^{(L)}(\mathbf{x}, \mathbf{z}_H^t, \mathbf{z}_L^t)$ is explicitly conditioned on the input embedding $\mathbf{x}$. In contrast, standard one-step feed-forward inference does not receive this additional conditioning on $\mathbf{x}$. We can say that *the policy*

*after reasoning steps behaves like a policy conditioned on the event that the answer is improved.* Under this interpretation, we can recover an optimality by Bayes' rule:

$$p(o = 1 \mid s, a) = \frac{p(a \mid s, o = 1)\, p(o = 1 \mid s)}{p(a \mid s)}.$$

Substituting $p(a \mid s, o = 1) \approx \pi_t^+(a \mid s_t)$ and $p(a \mid s) \approx \hat{\pi}_t(a \mid s_t)$ yields

$$p(o = 1 \mid s_t, a) \propto \frac{\pi_t^+(a \mid s_t)}{\hat{\pi}_t(a \mid s_t)}, \qquad (14)$$

where the proportionality constant $p(o = 1 \mid s_t)$ does not depend on $a$. Plugging (14) into the generic improvement step (9) gives

$$\pi_{t,w}(a \mid s_t) \propto \hat{\pi}_t(a \mid s_t) \left( \frac{\pi_t^+(a \mid s_t)}{\hat{\pi}_t(a \mid s_t)} \right)^{w} \qquad (15)$$

$$\propto \hat{\pi}_t(a \mid s_t)^{1-w}\, \pi_t^+(a \mid s_t)^{w}. \qquad (16)$$

Equation (16) is the policy-improvement analog of the *Bayes inversion* step: it rewrites a multiplicative optimality reweighting in terms of a *pair of policies* (unconditional vs. optimality-conditioned). Concretely, latent reasoning provides both factors $\hat{\pi}_t$ and $\pi_t^+$, and therefore implicitly defines the entire one-step improvement family $\{\pi_{t,w}\}_{w \geq 0}$. From (14), the improvement score is the log-ratio

$$A_t(s_t, a) := \log \pi_t^+(a \mid s_t) - \log \hat{\pi}_t(a \mid s_t) \qquad (17)$$
$$\equiv \log p(o = 1 \mid s_t, a) + \text{const}(s_t),$$

i.e., up to a state-dependent constant, the TRM step computes an *advantage-like* signal: it measures how much the reasoning step increases the relative probability of action $a$.

### 3.3. A supervised improvement criterion

For the prediction of the next-token, let $y^\star$ be the ground-truth token at the current position. Define the cross-entropy under the improved policy family (16):

$$L_t(w) := -\log \pi_{t,w}(y^\star \mid s_t).$$

Writing $\pi_{t,w}(\cdot \mid s_t) = \text{softmax}\big((1-w)\log \hat{\pi}_t + w \log \pi_t^+\big)$, a direct derivative gives

$$\frac{d}{dw} L_t(w) = \mathbb{E}_{a \sim \pi_{t,w}(\cdot \mid s_t)}\big[A_t(s_t, a)\big] - A_t(s_t, y^\star),$$

and

$$\frac{d^2}{dw^2} L_t(w) = \text{Var}_{a \sim \pi_{t,w}(\cdot \mid s_t)}\big[A_t(s_t, a)\big] \geq 0.$$

Thus, increasing $w$ (trusting the reasoning step more strongly) decreases the loss if and only if the ground-truth action has *above-average improvement score*:

$$A_t(s_t, y^\star) > \mathbb{E}_{a \sim \pi_{t,w}(\cdot \mid s_t)}\big[A_t(s_t, a)\big]. \qquad (18)$$

---

[1] For supervised next-token prediction, the simplest choice is $o = 1$ iff $a$ equals the ground-truth token; more general choices can encode task-level reward or self-consistency.

Equation (18) is a precise, testable condition for when a latent-reasoning update is helpful: the step improves prediction exactly when it preferentially increases the relative log-probability of the correct token compared to typical alternatives under the current policy.

### 3.4. Latent recursion as repeated policy improvement

Putting the pieces together, one TRM recursion step produces: (i) a reference policy $\hat{\pi}_t$, (ii) an approximate optimality-conditioned policy $\pi_t^+$, and therefore (iii) an implicit improvement model $p(o = 1 \mid s_t, a)$ via Bayes inversion (14). The family $\pi_{t,w}$ in (16) is exactly the corresponding policy-improvement update, with an advantage-like signal given by the log-ratio.

The recursion $t \mapsto t + 1$ can therefore be read as *iterating a learned policy improvement operator on a fixed input*: each latent step proposes a new distribution over actions and simultaneously provides the internal signal that justifies reweighting the previous distribution. The remaining question is not whether recursion converges, but whether training shapes $A_t(s, a)$ to be consistently aligned with task success—so that each step performs non-dead, directed improvement rather than redundant computation.

## 4. Deep Improvement Supervision

Section 3 reframes a latent-reasoning step as a *policy-improvement operator*: each recursion produces a *reference* policy $\hat{\pi}_t(\cdot \mid s_t)$ and a *post-reasoning* policy $\pi_t^+(\cdot \mid s_t)$, and their log-ratio $A_t(s_t, a) = \log \pi_t^+(a \mid s_t) - \log \hat{\pi}_t(a \mid s_t)$ acts as an advantage-like improvement signal. The analysis yielded a concrete *Advantage Margin* condition: a latent update helps exactly when the ground-truth action has an above-average improvement score under the interpolated policy family $\pi_{t,w}$ (Eq. (18)).

This gives a clean training target: rather than merely hoping that recursion learns to improve, we should *shape* the update so that, at every step, the log-ratio advantage prefers the next correct refinement over typical alternatives. Moreover, policies of the form $\pi \propto \hat{\pi} f(A^{\hat{\pi}})$ with $f$ non-negative and monotone are improvement operators. In our setting, $\pi_t^+$ already plays the role of an *optimality-conditioned* policy, and training should encourage $\pi_t^+$ to be the *next* improved distribution relative to $\hat{\pi}_t$.

**Deep Improvement Supervision (DIS)** operationalizes this idea by providing explicit *stepwise improvement targets* and supervising the *post-reasoning* readout at every recursion step so that each latent update realizes a directed improvement. Let $\mathbf{x}$ be the input and $\mathbf{y}^\star$ the ground-truth output sequence. Let $\Phi$ be a target generator that produces a sequence $\{\mathbf{y}_s^\dagger\}_{s=0}^{N_{\text{sup}}}$ with $\mathbf{y}_{N_{\text{sup}}}^\dagger = \mathbf{y}^\star$ and $\mathbf{y}_s^\dagger$ strictly closer to

$\mathbf{y}^\star$ than $\mathbf{y}_{s-1}^\dagger$ under a fixed discrepancy metric (e.g., expected Hamming distance).

We interpret the supervision step $s$ as enforcing a *single policy-improvement move*: the pre-reasoning policy $\hat{\pi}_s$ should behave like a policy centered on $\mathbf{y}_{s-1}^\dagger$, while the post-reasoning policy $\pi_s^+$ should behave like the improved (optimality-conditioned) policy centered on $\mathbf{y}_s^\dagger$. The most literal objective is therefore a *dual* cross-entropy:

$$\mathcal{L}_{\text{Dual}} = \sum_{s=1}^{N_{\text{sup}}} \Big[ \underbrace{\text{CE}(\ell_u^s, \mathbf{y}_{s-1}^\dagger)}_{\text{Anchor reference}} + \underbrace{\text{CE}(\ell_c^s, \mathbf{y}_s^\dagger)}_{\text{Supervise improvement}} \Big]. \quad (19)$$

In a recursive architecture, the input to step $s$ (which produces $\ell_u^s$) is a deterministic function of the post-reasoning state of the previous step (which produces $\ell_c^{s-1}$). Under teacher-forced recursion, training $\ell_c^{s-1}$ toward $\mathbf{y}_{s-1}^\dagger$ implicitly anchors $\ell_u^s$ near the previous target. Thus, we use the *single-loss*

$$\mathcal{L}_{\text{DIS}} = \sum_{s=1}^{N_{\text{sup}}} \text{CE}(\ell_c^s, \mathbf{y}_s^\dagger). \quad (20)$$

Inductively, minimizing Eq. (20) enforces the same stepwise anchoring as Eq. (19): forcing $\ell_c^{s-1} \to \mathbf{y}_{s-1}^\dagger$ ensures $\ell_u^s$ starts near $\mathbf{y}_{s-1}^\dagger$.

### 4.1. DIS as direct optimization of the Advantage Margin

Training $\ell_c$ on $\mathbf{y}_s^\dagger$ while $\ell_u$ is anchored to $\mathbf{y}_{s-1}^\dagger$ forces the residual $\Delta\ell^s = \ell_c^s - \ell_u^s$ encode the *direction of incremental improvement* from $\mathbf{y}_{s-1}^\dagger$ to $\mathbf{y}_s^\dagger$. Equivalently, in probability space, DIS shapes the log-ratio advantage $A_s(a) = \log \pi_s^+(a \mid s) - \log \hat{\pi}_s(a \mid s)$ assign higher-than-average advantage to the next target token(s), aligning with the theoretical Advantage Margin condition (Eq. (18)). This converts the previous section's existence/diagnostic statement ("improvement occurs iff the margin is positive") into a *constructive training rule*: make the margin positive at every step by supervising the post-reasoning policy on a strictly improving target sequence.

**Proposition 4.1** (Stepwise Advantage-Margin Alignment). *Assume that the target generator $\Phi$ provides strictly improving targets with respect to a fixed scoring distribution $P$, i.e., $\log \frac{P(\mathbf{y}_s^\dagger)}{P(\mathbf{y}_{s-1}^\dagger)} > 0$ for all $s$. Then minimizing the sequential loss $\mathcal{L}_{DIS}$ drives the expected Advantage Margin to be positive:*

$$\mathbb{E}\Big[\Delta\ell[\mathbf{y}^\star] - \mathbb{E}_{a \sim \pi_w} \Delta\ell[a]\Big] > 0,$$

*where the outer expectation is over the data (and model stochasticity) and the inner expectation is with respect to $\pi_w$ for any fixed $w \geq 0$.*

See Appendix C for more details. In short: standard training relies on the hope that recursion *happens* to learn a monotone $f(A)$-style reweighting. DIS enforces it by construction, by making each post-reasoning policy $\pi_s^+$ predict an explicitly improved target relative to the implicitly anchored reference policy $\hat{\pi}_s$. The result is a verifiable iterative refinement procedure in which each forward step is trained to behave like a concrete policy-improvement move.

## 4.2. Algorithm

Standard TRM training backpropagates a single terminal supervision signal through the entire recursive chain, so the model must simultaneously (i) discover useful intermediate states and (ii) align each latent update with improvement. DIS isolates the second requirement by *supervising the improvement operator itself*: each recursion step is trained to map the previous target $\mathbf{y}_{s-1}^{\dagger}$ to the next target $\mathbf{y}_s^{\dagger}$, directly instantiating the stepwise improvement from § 3.

Let $\{\mathbf{y}_s^{\dagger}\}_{s=1}^{N_{\text{sup}}}$ be constructed so that the expected discrepancy to $\mathbf{y}^{\star}$ strictly decreases with $s$. Let $\mathbf{x} \in \mathcal{X}$ be the input, $\mathbf{y}^{\star} \in \mathcal{Y}$ the final target, and $(\mathbf{y}, \mathbf{z})$ the TRM state passed across supervision steps. A *target generator* $\Phi$ produces a sequence of stepwise targets

$$\mathbf{y}_1^{\dagger}, \ \mathbf{y}_2^{\dagger}, \ \ldots, \ \mathbf{y}_{N_{\text{sup}}}^{\dagger} \ = \ \Phi(\mathbf{x}, \mathbf{y}^{\star}; \ 1{:}N_{\text{sup}}), \qquad (21)$$

with $\mathbf{y}_{N_{\text{sup}}}^{\dagger} = \mathbf{y}^{\star}$, such that the distance to the final solution decreases monotonically along the schedule under a task-appropriate discrepancy $d$ (e.g., Hamming distance over tokens). Each supervision step $s \in \{1, \ldots, N_{\text{sup}}\}$ receives a *puzzle embedding* $E(p)$.

Our framework admits diverse sources of intermediate targets $\Phi$ (any mechanism that yields monotone improvement targets is valid, since DIS only requires stepwise improvement, not a particular generative process):

1. **Programmable edits.** A deterministic generator creates a path from input to output, e.g. puzzle solvers that reveal a constant move in constraints per step, which yields $\mathbf{y}_{s+1}^{\dagger} = \text{Edit}(\mathbf{y}_s^{\dagger})$.

2. **LLM-generated plans.** A teacher LLM proposes intermediate sketches; these are projected onto the task's discrete output space to form $\{\mathbf{y}_s^{\dagger}\}$.

3. **Discrete corruption schedule.** Define a corruption process $q_{\beta}(\tilde{\mathbf{y}} \mid \mathbf{y}^{\star})$ (e.g., token masking or random replacement with rate $\beta$). Choose a decreasing noise schedule and sample $\mathbf{y}_s^{\dagger} \sim q_{\beta_s}(\cdot \mid \mathbf{y}^{\star})$ so that the targets become progressively less corrupted, yielding a simple monotone-improvement curriculum.[2]

---
[2]We do not position DIS as diffusion modeling: DIS does not

In our experiments, we use the simplest instantiation: stepwise targets via discrete corruption. Let $\mathbf{x}$ be the input and $\mathbf{y}^{\star}$ the ground-truth output. Choose a decreasing noise schedule $0 = \beta_{N_{\text{sup}}} < \cdots < \beta_2 < \beta_1 \leq 1$ and a token-level corruption kernel $q_{\beta}$ (e.g., masking at rate $\beta$). Define intermediate targets

$$\mathbf{y}_s^{\dagger} \ \sim \ q_{\beta_s}(\cdot \mid \mathbf{y}^{\star}), \qquad s = 1, \ldots, N_{\text{sup}}, \qquad (22)$$

so that $\mathbf{y}_{N_{\text{sup}}}^{\dagger} = \mathbf{y}^{\star}$ and, in expectation, the discrepancy with $\mathbf{y}^{\star}$ decreases with $s$. Please, see Appendix A for details.

During training, the model runs the TRM recursion for the $N_{\text{sup}}$ supervision steps. At each step $s$, it computes the post-reasoning logits $\ell_c^s$ and applies $\text{CE}(\ell_c^s, \mathbf{y}_s^{\dagger})$. By the theory in § 3, this stepwise supervision trains each latent update to increase the relative log-probability of the next improved target – i.e., to implement a non-trivial policy improvement operator rather than an unconstrained recurrent computation.

## 5. Experiments

In this section, we provide a detailed explanation and results on the complex N-Queens and ARC-AGI problems.

**Backbone.** Our DIS model reuses the *tiny single-network* TRM backbone but eliminates TRM's extra recursion and halting heads. We use a 2-layer Transformer block with RMSNorm, SwiGLU MLPs, and rotary position embeddings; weights are shared for both the latent-update and answer-update calls, exactly as in TRM's attention variant ("TRM-Att"), to isolate the contribution of DIS from capacity and architecture differences (Jolicoeur-Martineau, 2025). The task specific hyperparameters as the hidden layers size and reasoning steps are presented below, per task protocol. For additional experiments, see Appendices §A and §B.

From algorithmic perspective, we incorporate the time step $t$ into the model input and observe that an integer-based time index $(0, 1 \ldots, N_{sup})$ yields superior results compared to standard continuous diffusion time conditioning $t \in [0, 1]$). The total simplification that we made in comparison to the original TRM is as follows:

- **Recursion budget:** DIS: $T{=}1, n{=}2$ vs. TRM: $T{=}3, n{=}6$ The total step formula is $N_{sup}[T(n+1)]$, so TRM does $16 * [3 * (6 + 1)] = 336$, while we have $6(2 + 1) = \mathbf{18}$

- **Halting/ACT:** Although TRM/HRM uses halting, where HRM's ACT requires a second forward pass for the continue loss, DIS uses a fixed $N_{sup}{=}6$ without halting head and no extra forward pass.

---
require diffusion schedulers, score matching, or explicit density-ratio estimation (Lou et al., 2023). A corruption schedule is merely a convenient way to generate monotone intermediate targets in discrete spaces.

In our experiments, we match the hidden size of the TRM $D$=512 and call it the *medium model* in Table 1. When we use $D$=256 and the single decoder layer model, which results in 0.8 mil. parameters, we call it *compact*.

## 5.1. N-Queens

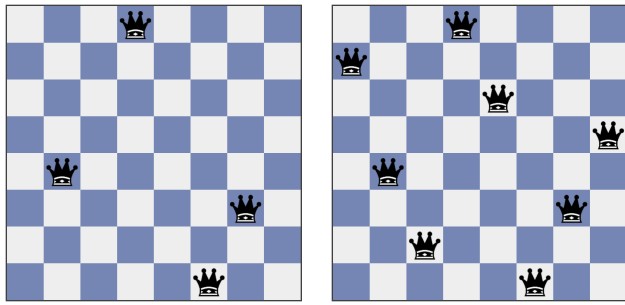

*Figure 1.* N-Queens reasoning problem example. Left is input and right is target solution.

**Task Format**. The N-Queens problem is a combinatorial reasoning task that involves placing $Q$ queens on a $8 \times 8$ chessboard (Oarga & Du, 2025). The fundamental objective is to arrange the queens such that no two queens threaten each other, which imposes the strict constraint that no two queens can share the same row, column, or diagonal.

This problem serves as a benchmark for evaluating a model's ability to generate valid configurations under complex constraints. The complexity of the task is directly determined by the parameter $Q$, which dictates the total number of queens that must be accommodated on the board. Complexity levels corresponding to problem instances ranging from $Q = 1$ to $Q = 7$ queens that are already on board, with lower values of $Q$ representing increasingly difficult reasoning challenges. In our experiments, we randomly sampled train and test experience with different complexity.

Only the classic $8 \times 8$ board and no augmentation was used, an example of input and target is represented in 1. We generated 7,200 train and 200 test examples with a sequence length of 64 and a vocabulary size of 3.

**Results**. This experiment demonstrates the impact of DIS

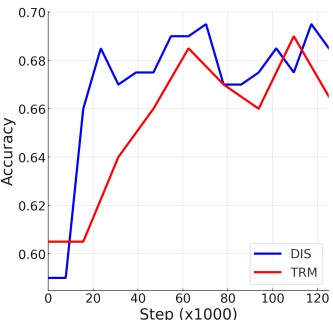

*Figure 2.* Accuracy curves on N-Queens problem.

training. As mentioned in §4, we used T = 1 and n = 2 for DIS, while the regular TRM of T = 3 and n = 6 was applied, with a halting head and 16 supervision steps. We achieved the same 0.69 accuracy while using much fewer inference steps. The architectures of the 0.8 mill parameters were used for both methods.

## 5.2. ARC Evaluation Protocol

**Task format.** ARC puzzles are sets of colored grids with 2–3 input–output demonstrations and 1–2 test inputs per task; the maximum grid size is $30 \times 30$. Accuracy is scored over all test grids with two attempts permitted per task (standard ARC scoring). We evaluate on the public evaluation sets of ARC-AGI-1 (800 tasks) and ARC-AGI-2 (1,120 tasks), following TRM.

**Data augmentation.** We adopt the augmentation pipeline of TRM: 1000 augmentations per puzzle via color permutations, dihedral-group transforms (90° rotations, flips/reflections) and translations. As in TRM, we also include the 160 ConceptARC tasks as additional training puzzles (Moskvichev et al., 2023). We attach a puzzle-specific embedding token per instance.

**Pre-/post-processing.** Inputs and outputs are tokenized as discrete color IDs; we concatenate demonstrations and the target input in the same sequence layout used by TRM, our positional scheme match theirs. For evaluation purposes, we apply the majority vote of the TRM to 1,000 augmented inferences per puzzle. Different number of passes are used and we report pass@2 as it is the most common setting.

## 5.3. ARC Model Settings

**Objective.** Each supervision step $s \in \{1, \ldots, 6\}$ is trained toward a *step-specific intermediate target* $\mathbf{y}_s^\dagger$ produced by a discrete corruption schedule of ground truth $\mathbf{y}^\star$ with monotonically decreasing noise. We use token-masking/replacement with a linearly decreasing mask rate over the 6 steps so that $\mathbb{E}[d(\mathbf{y}_s^\dagger, \mathbf{y}^\star)]$ decreases with $s$. The loss is the standard token-level cross-entropy in $f_O(\mathbf{y})$ against $\mathbf{y}_s^\dagger$.

**Optimization.** We follow the training recipe of TRM wherever applicable: Adam-Atan with $\beta_1$=0.9, $\beta_2$=0.95, a 2k warm-up and the stable-max cross-entropy variant for stability. For ARC experiments, we use weight decay 0.1 and we did not find EMA important. Also, we match batch sizing; embedding LR warm-up and an elevated embedding LR (as in TRM) are retained.

**Deep improvement supervision loop.** For each mini-batch we run $N_{\text{sup}}$=6 DIS steps. At each step, we execute a single external cycle (since $T$=1) comprising two internal latent/answer updates ($n$=2), backpropagating throughout the cycle; then we detach $(\mathbf{y}, \mathbf{z})$ before the next step. We do

not train a halting/ACT head. Importantly, when using a discrete diffusion model, the supervision trajectories are generated/sampled on the fly, as in a regular diffusion process. Therefore, for the same task, *we can have various diffusion steps* towards the target.

**Test-time compute.** We run the same $N_{sup}=6$ steps at evaluation. To compare fairly with previous `ARC` protocols, we keep TRM's test-time augmentation vote: run the model over 1000 geometric/color augmentations of a puzzle and return the most common prediction.

### 5.4. `ARC` Results

For our experiments, we replicated the TRM experiments and achieved slightly lower results than those reported in the original paper. We also re-implemented TRM with the same hyperparameter settings as in our *medium* model to compare the methods with identical resources, we set $T = 1$, $n = 2$, but still use $N_{sup}=16$ for TRM, because the halting mechanism remained active. In addition, we implemented a smaller network to reproduce a compact model consisting of only 0.8 million parameters.

*Table 1.* Model Performance Comparison, pass@2

| Method | Params | ARC-1 | ARC-2 |
|---|---|---|---|
| **Chain-of-thought, pretrained** | | | |
| Deepseek R1 | 671B | 15.8 | 1.3 |
| Claude 3.7 16K | ? | 28.6 | 0.7 |
| o3-mini-high | ? | 34.5 | 3.0 |
| Gemini 2.5 Pro 32K | ? | 37.0 | 4.9 |
| Grok-4-thinking | 1.7T | 66.7 | 16.0 |
| Bespoke (Grok-4) | 1.7T | **79.6** | **29.4** |
| **Small-sample training** | | | |
| TRM-compact | 0.8M | 12.0 | 0.0 |
| DIS-compact (Ours) | 0.8M | 24.0 | 0.0 |
| TRM-medium | 7M | 27.1 | 0.0 |
| DIS-medium (Ours) | 7M | 40.0 | 2.5 |
| TRM | 7M | 40.4 | 3.3 |
| DIS | 7M | 41.3 | 6.0 |

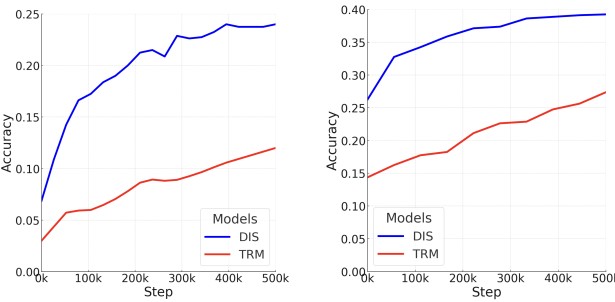

*Figure 3.* The DIS and TRM models pass@2 scores under the compact (left) and medium (right) setups.

The results are presented in Table 1 and Figure 3. As shown, for the compact model we dramatically outperform the orig-

inal TRM. This shows that for TRM, latent reasoning steps are important. We reduced the total number of latent steps nine times and achieved a significant improvement in performance. However, explicit supervision of each step can overcome this drawback by simplifying the task for the TRM, meaning that longer latent reasoning is unnecessary. Furthermore, our medium model outperforms the medium TRM and achieves results comparable to the original TRM.

**Shaped credit assignment across supervision steps.** In baseline TRM, every step is trained directly against $\mathbf{y}^\star$, leaving it to the model to discover a self-improvement curriculum DIS supplies *explicit* intermediate targets $\{\mathbf{y}_s^\dagger\}$, aligning the step-$s$ gradients with a concrete improvement.

This reduces the burden on the latent state $\mathbf{z}$ to implicitly encode a stepwise plan and can accelerate optimization in scarce-data regimes, where TRM has been shown to be the most effective. DIS retains TRM's minimal two-feature interface $(\mathbf{y}, \mathbf{z})$, single tiny network reused for both updates, and the schedule of $T-1$ no-grad cycles followed by one grad cycle. It inherits the simplicity advantages of TRM while changing only the supervision signal.

**Compute and stability.** With a monotone schedule, DIS turns each supervision step into a measurable sub-goal. We preserve the TRM's compute profile per step (one gradient-bearing recursion cycle) and avoid HRM/TRM-style ACT. If targets are generated offline, the runtime overhead is negligible; if produced online (e.g., by a teacher model), they can be cached or amortized across epochs. For training we used the same 4 `GPU H100` setting as TRM, but learning takes $\approx 40$ hours against 72 in TRM.

## 6. Conclusion

We demonstrate that small iterative reasoning models can achieve competitive performance on complex reasoning tasks such as the ARC. By reinterpreting TRMs through the lens of reinforcement learning, we reveal that TRMs implicitly perform policy improvement, where a latent *working memory* state guides the model toward better solutions over recursive steps. The key contribution is Deep Improvement Supervision, builds on this insight by introducing a structured, stepwise training regime. DIS provides intermediate targets through a discrete diffusion process (Ho & Salimans, 2022; Frans et al., 2025), transforming the challenging problem of long-term credit assignment into a more tractable supervised learning task. Our approach not only simplifies training but also enhances efficiency, reducing the number of forward passes by 18x with high accuracy.

## Impact Statement

This paper presents work whose goal is to advance the field of Machine Learning. There are many potential societal consequences of our work, none which we feel must be specifically highlighted here.

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

# A. Illustrations

```
def latent_reasoning(x, y, z, n=2):          1
    with torch.no_grad():                    2
        for j in range(T-1):                 3
            for i in range(n):               4
                z = net(x, y, z)             5
            y = net(y, z)                    6
    for i in range(n):                       7
        z = net(x, y, z)                     8
    y = net(y, z)                            9
    return (y.detach(), z.detach()),         10
        output_head(y)
                                             11
# Deep Improvement Supervision               12
for x_input, y_true in train_dataloader:     13
    y, z = y.init, z.init                    14
    for step in range(N_supervision):        15
        y_step = f(x_true, y_true, step)     16
        x = input_embedding(x_input, step)   17
        (y, z), y_hat = latent_reasoning(x   18
            , y, z)
        loss = softmax_cross_entropy(y_hat   19
            , y_step)
        loss.backward()                      20
        opt.step()                           21
        opt.zero_grad()                      22
```

*Figure 4.* Pseudocode for reasoning with deep improvement supervision. With $T = 1$ (as in our *medium* settings), **we avoid the large (no-grad) cycle** and significantly reduce computational time.

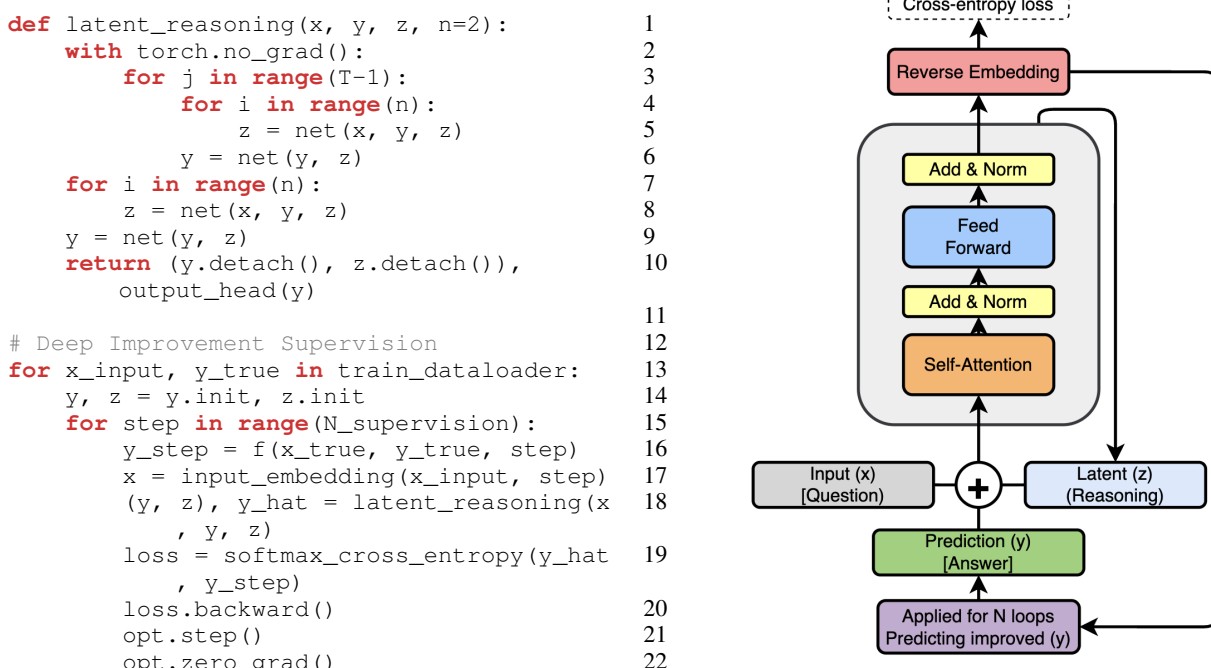

*Figure 5.* DIS model architecture. Algorithm starts with the embedded input question **x**, initial embedded answer **y**, and latent state $z$. For up to $n$ improvement steps, it tries to improve its answer **y** by simulating a discrete diffusion process, addressing any errors from its previous answer in an parameter-efficient manner.

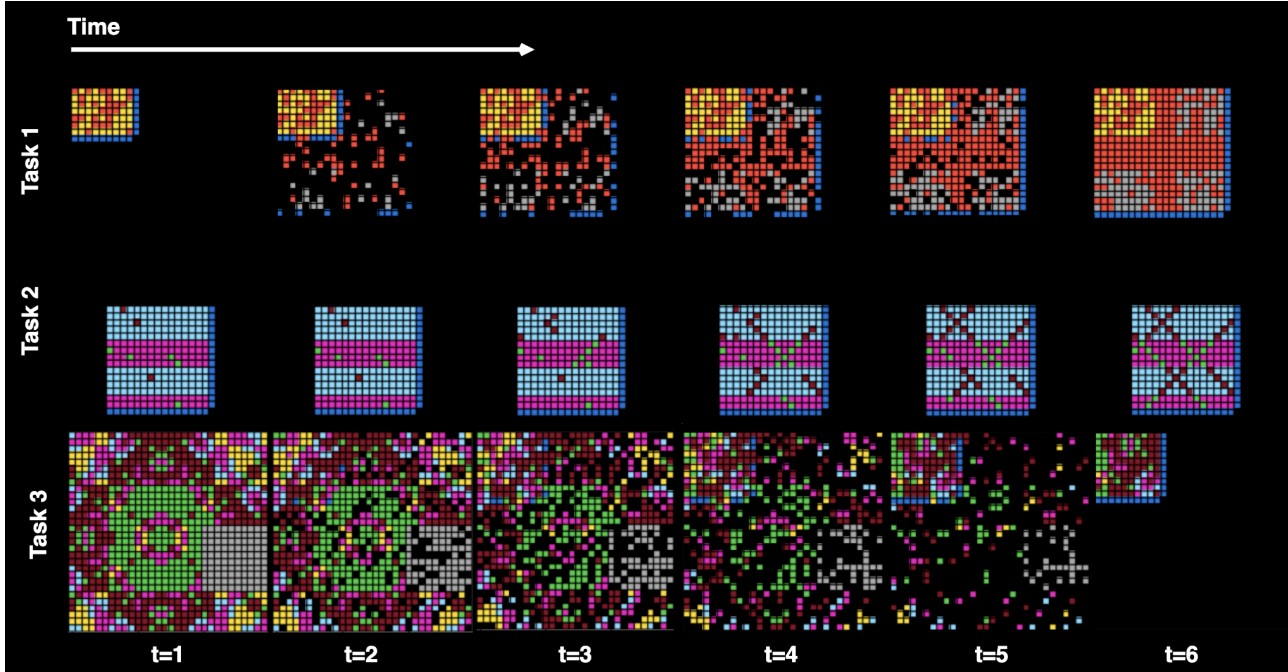

*Figure 6.* ARC-1 Solution examples. The corruption process is shown over six steps, from the initial input at time $t = 0$ to the target at time $t = 6$. A single training sample is illustrated per task.

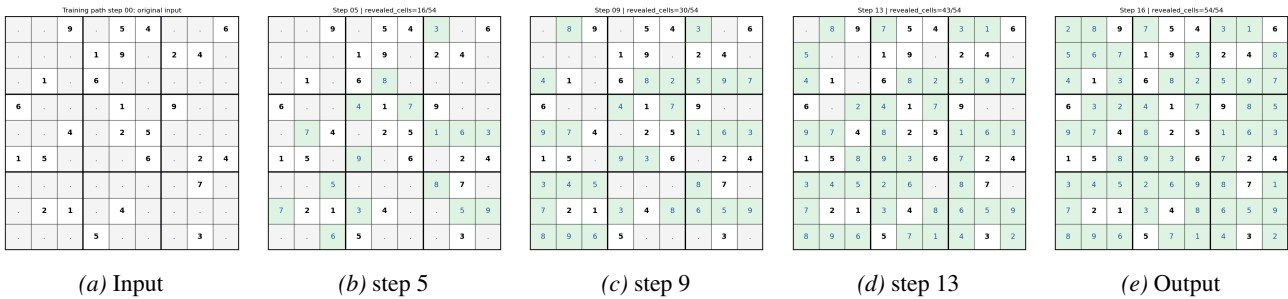

*Figure 8.* Sudoku solution example.

## B. Sudoku Experiments with an MLP Backbone

We evaluate Deep Improvement Supervision (DIS) on Sudoku using a lightweight MLP backbone. Each puzzle is represented as an $81$-token sequence over digits $0, \ldots, 9$, where $0$ denotes an empty cell. The model keeps the same recurrent answer and latent states $(y, z)$ as TRM, but replaces the attention block with a residual MLP update network. This setting isolates the effect of the supervision strategy from the architectural benefits of attention.

For a fair comparison, our model uses the same compute budget as TRM, but does not use a halting head. Instead, we run a fixed number of supervision and reasoning steps and train each step with Deep Improvement Supervision. Intermediate targets $y_s^{\dagger} * s = 1^{N*\sup}$ are generated from the solved grid $y^{\star}$ using a monotone discrete corruption schedule, with $y_{N_{\sup}}^{\dagger} = y^{\star}$. The training objective is $\mathcal{L}_{\mathrm{DIS}} = \sum_{s=1}^{N_{\sup}} \mathrm{CE!} \left( f_O(y_s), y_s^{\dagger} \right).$

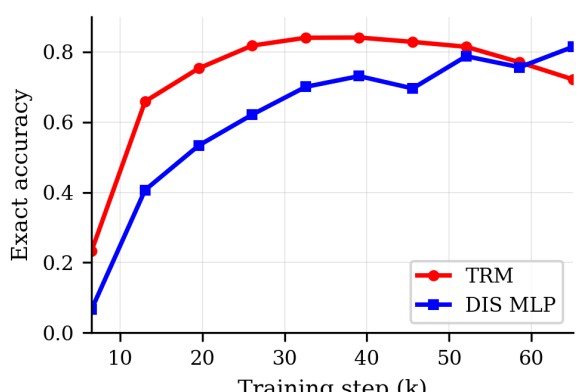

*Figure 7.* Sudoku exact accuracy for MLP backbones under the same compute budget. Our DIS model does not use a halting head and reaches a comparable final score, and learns more slowly than the TRM baseline. Understanding this slower optimization on Sudoku remains an important direction for future work.

Under this matched budget, the MLP-DIS model achieves the same final Sudoku score as TRM despite not relying on adaptive halting. However, we observe that learning is slower: the model requires more optimization steps to reach the same level of performance. This suggests that, unlike in ARC and N-Queens, the stepwise corruption targets used by DIS may not provide the most effective improvement trajectory for Sudoku. Understanding why DIS does not outperform TRM on this task, and whether Sudoku requires task-specific intermediate targets or adaptive computation, remains an important direction for future study. Please see figures above.

## C. Proofs

*Proof of Proposition 4.1.* Fix a supervision step $s$ and its state $s_s$ (teacher-forced), with previous target $\mathbf{y}_{s-1}^{\dagger}$ and current target $\mathbf{y}_s^{\dagger}$. Write the pre/post policies as

$$\hat{\pi}_s(\cdot \mid s_s) = \mathrm{softmax}(\ell_u^s), \qquad \pi_s^+(\cdot \mid s_s) = \mathrm{softmax}(\ell_c^s),$$

and define the logit residual $\Delta \ell^s(a) = \ell_c^s[a] - \ell_u^s[a]$.

**Step 1: Relate the margin in logits to the margin in advantages.** Define the log-ratio advantage (Section 3)

$$A_s(a) := \log \pi_s^+(a \mid s_s) - \log \hat{\pi}_s(a \mid s_s).$$

Using $\log \mathrm{softmax}(\ell)[a] = \ell[a] - \log \sum_{a'} e^{\ell[a']}$, we have

$$A_s(a) = \left( \ell_c^s[a] - \ell_u^s[a] \right) - \left( \log \mathbf{z}_c^s - \log \mathbf{z}_u^s \right) = \Delta \ell^s(a) - C_s,$$

where $\mathbf{z}_c^s = \sum_{a'} e^{\ell_c^s[a']}$, $\mathbf{z}_u^s = \sum_{a'} e^{\ell_u^s[a']}$, and $C_s$ does not depend on $a$. Therefore, for any distribution $\mu$ over actions,

$$\Delta\ell^s(\mathbf{y}_s^\dagger) - \mathbb{E}_{a\sim\mu}\Delta\ell^s(a) \;=\; A_s(\mathbf{y}_s^\dagger) - \mathbb{E}_{a\sim\mu}A_s(a).$$

So it suffices to prove the claim for $A_s$.

**Step 2: DIS makes the post-reasoning policy concentrate on the improved target.** DIS minimizes $\mathrm{CE}(\ell_c^s, \mathbf{y}_s^\dagger) = -\log \pi_s^+(\mathbf{y}_s^\dagger \mid s_s)$ at every step, so along training we drive

$$\pi_s^+(\mathbf{y}_s^\dagger \mid s_s) \;\to\; 1.$$

Equivalently, for any $\varepsilon > 0$ there exists a training time after which $\pi_s^+(\mathbf{y}_s^\dagger \mid s_s) \geq 1 - \varepsilon$.

**Step 3: The interpolated policy $\pi_{s,w}$ also concentrates on the improved target.** Recall $\pi_{s,w}(a \mid s_s) \propto \hat{\pi}_s(a \mid s_s)^{1-w}\pi_s^+(a \mid s_s)^w$ for any fixed $w \geq 0$. Since softmax policies have full support, $\hat{\pi}_s(\mathbf{y}_s^\dagger \mid s_s) > 0$. Hence the multiplicative form implies: if $\pi_s^+(\mathbf{y}_s^\dagger \mid s_s) \geq 1 - \varepsilon$, then also

$$\pi_{s,w}(\mathbf{y}_s^\dagger \mid s_s) \;\geq\; 1 - \varepsilon_w \qquad \text{with } \varepsilon_w \to 0 \text{ as } \varepsilon \to 0$$

(for fixed $w$, because $\pi_{s,w}$ inherits concentration from $\pi_s^+$).

**Step 4: Concentration implies a positive advantage margin.** Decompose the expectation:

$$\mathbb{E}_{a\sim\pi_{s,w}}A_s(a) = \pi_{s,w}(\mathbf{y}_s^\dagger)\,A_s(\mathbf{y}_s^\dagger) + \sum_{a\neq\mathbf{y}_s^\dagger}\pi_{s,w}(a)\,A_s(a).$$

Thus

$$A_s(\mathbf{y}_s^\dagger) - \mathbb{E}_{a\sim\pi_{s,w}}A_s(a) = \sum_{a\neq\mathbf{y}_s^\dagger}\pi_{s,w}(a)\Big(A_s(\mathbf{y}_s^\dagger) - A_s(a)\Big).$$

Under DIS, $\pi_s^+(\mathbf{y}_s^\dagger) \to 1$ forces $\log\pi_s^+(\mathbf{y}_s^\dagger) - \log\pi_s^+(a) \to +\infty$ for $a \neq \mathbf{y}_s^\dagger$, while $\hat{\pi}_s$ is fixed at that training step; hence $A_s(\mathbf{y}_s^\dagger) - A_s(a) > 0$ for all $a \neq \mathbf{y}_s^\dagger$ once training is sufficiently advanced. Since $\pi_{s,w}(a) \geq 0$ and $\sum_{a\neq\mathbf{y}_s^\dagger}\pi_{s,w}(a) = 1 - \pi_{s,w}(\mathbf{y}_s^\dagger) = \varepsilon_w$, we obtain

$$A_s(\mathbf{y}_s^\dagger) - \mathbb{E}_{a\sim\pi_{s,w}}A_s(a) \;>\; 0$$

as soon as $\varepsilon_w$ is small enough and the strict gaps $A_s(\mathbf{y}_s^\dagger) - A_s(a)$ are positive.

**Step 5: Apply at the final target and take expectation.** At the last supervision step $s = N_{\mathrm{sup}}$, $\mathbf{y}_s^\dagger = \mathbf{y}^\star$. Combining Steps 1–4 and taking expectation over data and model stochasticity yields

$$\mathbb{E}\Big[\Delta\ell[\mathbf{y}^\star] - \mathbb{E}_{a\sim\pi_w}\Delta\ell[a]\Big] = \mathbb{E}\Big[A(\mathbf{y}^\star) - \mathbb{E}_{a\sim\pi_w}A(a)\Big] \;>\; 0.$$

Finally, the assumption $\log\frac{P(\mathbf{y}_s^\dagger)}{P(\mathbf{y}_{s-1}^\dagger)} > 0$ ensures the supervision sequence corresponds to genuine stepwise improvements under the task scoring $P$, i.e., the desired direction of advantage is consistent with the target generator rather than arbitrary. $\qquad\square$

## D. Discussion

**Potential algorithmic improvements.** A key limitation of the current implementation is the use of a fixed number of supervision steps for every `ARC` task. However, task complexity varies significantly; some tasks may benefit from a higher number of denoising steps, while others require fewer. This observation aligns with findings from the original TRM paper, which highlighted the significant contribution of the halting mechanism to the final performance. Therefore, explicitly predicting the necessary number of denoising steps for each task could potentially improve overall model efficiency and accuracy.

Another promising direction for technical improvement involves the adoption of a discrete latent space. This approach has been successfully utilized in deep learning architectures such as the Dreamer model (Hafner et al., 2019) and VQ-VAE (Razavi et al., 2019), where latent spaces have proven to be robust and scalable for generative tasks.

**Alternative Improvement Generators.** As described in §4.2, there are several viable methods to generate intermediate steps. Although the prior discrete diffusion serves as the main source in this work, our framework is designed to support various step-by-step approaches.

We also investigated the use of LLM-generated trajectories between transition samples and their targets, specifically utilizing the Gemini 2.5 Pro model. We trained a compact network (0.8 million parameters) on these trajectories; however, this method underperformed compared to the diffusion prior.

We hypothesize that LLM-generated trajectories fail to provide a monotonic improvement path, often introducing highly nonlinear "jumps" between intermediate steps that are difficult for a small model to capture.

Consequently, the model trained with LLM improvement supervision achieved only 10% accuracy, compared to the 24% achieved with the diffusion prior. Exploring code-based generation of intermediate steps remains a promising direction for future work to improve the algorithm's performance.

