# OpenReview forum: "Latent Reasoning in TRMs is Secretly a Policy Improvement Operator"
_ICML.cc/2026/Conference — ICML 2026 regular_

### Official Review · Reviewer_GnRY · 2026-02-24

**Soundness:** 2
**Presentation:** 2
**Significance:** 1
**Originality:** 2
**Overall Recommendation:** 3
**Confidence:** 2

**Summary:**

The paper offers an insightful contribution to the understanding of recursive reasoning and policy improvement, providing a method that enhances the training and efficiency of small reasoning models.

**Compliance With Llm Reviewing Policy:**

Affirmed.

**Final Justification:**

The paper provides an insightful contribution to understanding recursive reasoning and policy improvement, introducing a method that enhances the training and efficiency of small reasoning models. During the rebuttal phase, the authors addressed many of my questions, leading me to raise my score. However, the literature review remains limited, the theoretical analysis does not fully explain why the method works, and the experimental evaluation is somewhat narrow. For these reasons, I give a weak reject rather than an accept.

**Key Questions For Authors:**

1.Could you provide visualizations, such as improvement curves or feature importance plots, to show how the DIS method improves the reasoning process over iterations and the specific steps where improvements are most noticeable?

2.DIS demonstrate the effectiveness of DIS on ARC-AGI and N-Queens. Are there other reasoning tasks or datasets that you believe would benefit from your approach, and if so, could you outline how the model might perform across different domains or problem types?

3.Could you provide an ablation study to break down the individual contributions of key components in the DIS method, such as the role of the supervision scheme or the recursive steps, in improving model performance?

**Limitations:**

1.The authors should include a more detailed discussion of the limitations of their approach, particularly regarding its generalizability, computational complexity, and reliance on fixed supervision steps.

2.The paper references a relatively small set of papers. A more extensive literature review, covering a broader range of recent work on recursive reasoning models, policy improvement algorithms, and related techniques, would help contextualize the proposed method better and highlight its contributions more clearly.

**Strengths And Weaknesses:**

**Strengths**

1.The paper provides a fresh view on latent reasoning by connecting it to policy improvement algorithms, thus enhancing our understanding of recursive reasoning in small models.

2.DIS addresses the challenge of training recursive models by providing structured supervision at each step, improving training stability and model efficiency.

**Weaknesses**

1.The paper does not sufficiently compare the proposed method with other SOTA models in recursive reasoning or latent reasoning. A more thorough comparison with existing techniques could help highlight the strengths and weaknesses of the approach.

2.The paper only presents experiments on the N-Queens and ARC-AGI tasks. A broader range of experiments across various domains, such as math or code reasoning tasks, would provide a more comprehensive evaluation of the proposed method's generalizability and robustness.

3.The paper lacks more in-depth analysis, such as visualizations or detailed results that would help illustrate the importance of the DIS method. Visual aids, such as improvement curves, heatmaps, or feature importance plots, could provide a clearer understanding of how DIS contributes to performance improvements.

4.The paper does not provide sufficient explanations for why specific experimental configurations, such as the number of supervision steps or the choice of parameters, were selected. Understanding the rationale behind these decisions is crucial for assessing the robustness of the method.

5.The paper lacks an ablation study to demonstrate the individual contributions of different components of the DIS method. Additionally, a parameter study would help understand the sensitivity of the model's performance to different hyper-parameter settings.

---

> ### Author Rebuttal · Authors · 2026-03-30
>
> #### **Answers to Weaknesses**
>
> 1. **Compare with other SOTA models in recursive reasoning or latent reasoning.**
>
>     Dear Reviewer, HRM and TRM are SOTA in recursive reasoning or latent reasoning. There are other latent reasoning mechanisms, but they belong to a different deep learning paradigm, mostly the autoregressive one. As you can see in the original TRM and HRM papers, the positioning of these (fixed-point equilibrium) approaches is isolated from other methods, largely because other methods are not designed to work with a small number of parameters.
>
>     To support our statement, we conducted experiments using the COCONUT training approach, which is SOTA in latent reasoning using autoregressive models. On both the 7m and 0.8m parameter models, this approach achieved 0.0 accuracy on the ARC-1 problem. We understand that these approaches require additional careful design to be applicable to combinatorial reasoning tasks, which is another reason we did not compare with them. We added discussion on these results in the revised version.
>
> 2. **Experiments across various domains, such as math or code reasoning tasks...**
>
>     Dear Reviewer, we appreciate your suggestion, but evaluating on standard autoregressive tasks (e.g., code generation) introduces external constraints and falls outside the scope of recursive latent-state mechanics. TRM and HRM, and consequently DIS, are not designed for these kinds of tasks. Extending these approaches to such tasks is out of the scope of our paper.
>
> 3. **The paper lacks more in-depth analysis, such as visualizations that would help illustrate the importance of the DIS method...**
>
>      Thank you for your suggestion, we will includes visualizations detailing the evolution of prediction quality across recursive steps. Our new figure showing comparison of forward residuals and PCA trajectories. It shows that the $Z_H$-module steadily converges, while the $Z_L$-module repeatedly converges within cycles before being reset by $Z_H$, supporting our statement on considering $Z_L$ module as optimality condition. This figure is motivated by Figure 3 from HRM paper. The learned trajectories during the solution to the ARC problem are also will be included.
>
> 4. +5. **Why specific experimental configurations, such as the number of supervision steps or the choice of parameters, were selected.**
>
>      Dear reviewer, our initial choices are made to be as close as possible to original TRM + test our method with significantly decreased number of forward steps (18x less). We conducted ablation study to isolate effect of each step. Please see our ablation study on 0.8M model for ARC-1 dataset below
>
>     | Model | Halting | Superv. | Recursion | Input Index | Score |
>     | :--- | :---: | :---: | :---: | :---: | :---: |
>     | **DIS** | **No** | **6** | **3** | **Yes** | **24** |
>     | DIS | Yes | 6 | 3 | Yes | 24 |
>     | DIS | No | 6 | 3 | No | 23 (-1) |
>     | DIS | Yes | 16 | 18 | Yes | 26 (+2) |
>     | **TRM** | **Yes** | **16** | **18** | **No** | **12** |
>     | TRM | Yes | 16 | 18 | Yes | 12 |
>     | TRM | Yes | 6 | 3 | Yes | 7 |
>
>     The **bolded lines** in the table represent the default settings reported in the paper.  Our results show that the halting mechanism provides no additional gains when using a small number of supervision steps (6). In contrast, the supervision rule remains highly robust with a score of 24 even without halting, while the TRM model's performance collapses significantly without halting. Increasing the number of latent recursion steps acts as the strongest performance driver for both architectures, even though we reported our results on settings with 18 times fewer forward passes for TRM.
>
>
> #### **Key Answers For Reviewer:**
> 1. See answer 3.
> 2. Answer 1.
> 3. Answer 4+5.
>
> #### **Limitations**
>
> 1. *Detailed discussion of the limitations.*
>
>     We expanded the Limitations section to formalize the boundaries of our framework:
>     * **Generality:** Empirical validation is currently scoped to TRM-style recursive latent reasoning paradigms rather than ubiquitous autoregressive modeling.
>     * **Fixed Supervision Depth:** While algorithmically stable, imposing a fixed sequence of intermediate targets ($N_{sup}=6$) may prove suboptimal for tasks of varying complexity. Developing adaptive depth mechanisms remains a critical direction for future work.
>     * **Computational Trade-offs:** Generating monotonic intermediate targets introduces specific upstream design prerequisites, although runtime overhead remains negligible when targets are generated offline and cached.
>
> 2. *A more extensive literature review...*
>
>     We thank the reviewer for this sugesstion. We will substantially broadened the literature review.
>
> #### **Concluding remarks**
>
> Thank you for your thoughtful suggestions. We would appreciate it if you could let us know whether our clarifications address your concerns. If there are any remaining issues, we would be happy to discuss them further.

---

> > ### Author Rebuttal · Reviewer_GnRY · 2026-04-01
> >
> > Thank you for your detailed response. I appreciate the clarifications provided.
> >
> > I do have a follow-up question:
> >
> > 1. Regarding Answer 1 and baseline validity: The authors report that the SOTA latent reasoning method (COCONUT) achieved 0.0 accuracy on ARC-1. This is a highly unusual result for a SOTA method and suggests a potential misalignment in how the baseline was adapted. Without a more successful or properly tuned baseline comparison, how can we be sure that the performance gain isn't simply due to the DIS model being the only one specifically "overfitted" to the recursive requirements and unique data augmentations of the ARC evaluation protocol? Did the authors test a version of TRM supervised at every step with the final ground-truth target to isolate the effect of the "improvement" schedule from the effect of dense supervision?
> >
> > 2. Regarding Answer 2 and the scope of the theory: The authors claim to investigate an important aspect of latent reasoning as a fundamental operator, yet the response dismisses standard reasoning benchmarks (like math or code) as "outside the scope" because they are autoregressive. If this "secret" policy improvement operator is truly a core mechanism of latent recursion, why would its validity be restricted to the specific architectural quirks of TRM/HRM or the visual-grid constraints of ARC?
> >
> > 3. Regarding the promised visualizations in Answer 3, could the authors provide a result similar to Figure 7 in the HRM paper?  This paper strives to consider the concept of latent reasoning through a policy improvement lens; therefore, a plot showing the evolution of prediction certainty or latent trajectories across recursion steps is essential.
> >
> > 4. Regarding the promised visualizations in Answer 4&5, while the table shows the performance under different settings, it does not fully address my question regarding the rationale behind the design choices. Specifically, I am interested in understanding why each component (halting mechanism, supervision scheme, recursion steps, input indexing) was selected in the default DIS configuration, and how much each individually contributes to the overall performance.
> >
> > 5. While the DIS model outperforms the TRM, it also provides the model with dense, intermediate supervised targets which the original TRM does not receive. The paper strives to consider the concept of a "Policy Improvement Operator," but how much of the success is actually due to this theoretical framework versus simply the benefit of more frequent supervised signals?
> >
> > Overall, I think the proposed method represents a simple code modification and a small trick that seems effective in practice. However, the literature review is not thorough, the theoretical analysis does not deeply explain why the method works, and the experimental evaluation is limited, and comparisons are made only against TRM-based methods.

---

> > > ### Author Response · Authors · 2026-04-01
> > >
> > > Thank you for your questions and for the opportunity to clarify our scope and method. Let us start by answering the second question, because we find it important to clarify our scope first.
> > >
> > > **2. Scope of Latent Reasoning**
> > > We apologize for this confusion. In our analysis, "latent reasoning" refers specifically to the mechanisms proposed in the TRM/HRM (Tiny Recursive Model/Hierarchical Recursive Model) frameworks, rather than to all general types of latent reasoning. Our paper aims to provide an interpretation of how **tiny** recursive models can perform so efficiently and how latent recursions facilitate this performance. These models perform well, but an interpretation of why this is the case has not been provided, we are closing this gap and providing ways for improvement. To prevent further confusion, we will clarify this even more explicitly in our abstract and introduction, and we will add a related work section for clear positioning. We are ready to change our title to 'Your latent recursive reasoning is secretly a policy improvement operator' if you find it describes the scope of the paper better.
> > >
> > > **1. Comparison**
> > > Dear reviewer, there is no misalignment in how the baseline was adapted. The COCONUT [1] is a method for reasoning in latent space designed for large language models with over a billion parameters. In contrast, our method belongs to the TRM family, "tiny" models with a maximum tested size of 7M parameters. COCONUT's latent reasoning mechanism was not designed for tiny models, and method does not downscale effectively to a few million parameters. This result highlights that TRM/HRM reasoning methods are unique, and **we are focusing only on this type of latent reasoning**. We believe developing these efficient, alternative small models is vital for the deep learning community, and our results can provide a step toward developing a such models for code and math reasoning tasks.
> > >
> > > Regarding the use of supervised signals at every step with final ground-truth labels: **this is exactly what the original TRM approach does** (please see Algorithm 1 in the original TRM paper). We compared DIS and TRM in 1:1 settings, where the only difference is the way supervision looks. See the ablation table, especially the first and last lines.
> > >
> > > [1] https://arxiv.org/pdf/2412.06769
> > >
> > > **3. Visual Comparisons**
> > > Thank you for this suggestion. We will certainly include these figures as well. Please find examples of such visualizations in the provided anonymous Google Drive folder: https://drive.google.com/drive/folders/1XKrVf5HVa7eeeJUfcoBt8zpNfKVcsJlC?usp=sharing. We will update them to be arranged similarly to Figure 7 and provide a 1:1 comparison with HRM using the exact same puzzles, including Sudoku.
> > >
> > > **4. Clarification of Contributions and Rationale**
> > > Our goal was to show that we can strip away unnecessary details if our training procedure is different. As shown in the ablation table, the most significant performance gain comes from the proposed deep improvement supervision, not the removal of components. In fact, our method performs better without these removals, but removing components without our method causes performance to drop significantly. We remove these details because the general goal of the tiny recursive models paradigm is to provide as compact a deep learning reasoning architecture as possible.
> > >
> > > **5. Policy Improvement and Supervised Signals**
> > > Dear reviewer, the number of signals is the same for our model and TRM, the only difference is the signal itself. Our policy improvement-based interpretation holds true not only for our method (DIS) but also for the TRM and HRM architectures (see Sections 3.1 and 3.2). What we did is provide the theoretical interpretation for TRM and HRM and then, acknowledging the importance of the monotone signal for correct policy improvement, provide a practical framework to achieve improved results. Without our theory, it is unclear why a monotone improvement schedule of the targets works better.

---

### Official Review · Reviewer_eHog · 2026-03-12

**Soundness:** 2
**Presentation:** 3
**Significance:** 3
**Originality:** 3
**Overall Recommendation:** 4
**Confidence:** 4

**Summary:**

This paper argues that many latent-recursion steps are dead compute rather than effective depth. It formalizes one recursion step as an implicit policy improvement over next-token distributions, using a product-policy view and an advantage-like log-ratio. This yields a step-level criterion intended to predict when a step meaningfully improves the distribution. Based on this view, the method Deep Improvement Supervision adds monotonic intermediate targets via discrete corruption schedules. Reported results show fewer recursion steps with comparable or better performance on N-Queens and ARC-AGI.

**Compliance With Llm Reviewing Policy:**

Affirmed.

**Final Justification:**

I thank the authors for their detailed rebuttal. I'll maintain my positive score of 4.

**Key Questions For Authors:**

Does the proposed step-level margin reliably predict useful steps across tasks and model sizes, and can it detect dead compute steps in deployment settings?

How does the method define improvement targets for tasks with many valid outputs, and how sensitive is it to non-monotonic intermediate trajectories?

**Limitations:**

Yes

**Strengths And Weaknesses:**

### Strengths

The paper provides a clean and reusable formalization of latent recursion as a policy-improvement-like update over token distributions, including an advantage-style signal and a testable step-level condition (Eq. 18).

The proposed training objective (DIS) is well aligned with the analysis: stepwise supervision explicitly encourages each recursion step to be directionally useful rather than relying on weak end-to-end credit assignment.

The “dead compute” perspective is practically relevant for recursive reasoning models: it targets both efficiency and training stability, and helps operationalize what it means for a recursion step to be “useful.”

The connection between latent recursion, product-policy updates, and guidance-style interpretations is a nice conceptual bridge; DIS is a reasonably novel instantiation that turns this analysis into a training recipe.


### Weaknesses

**Lack of analysis experiments** As far as I see, current experiments do not directly demonstrate that the proposed advantage-margin criterion reliably predicts when a recursion step is beneficial. For example, the paper could report the distribution of the margin across steps, correlate it with stepwise accuracy/loss improvements, or use it to identify dead-compute steps in practice.


**Generalization to open-domain tasks.** It's currently unclear how DIS works when targets are ambiguous, multi-solution, or open-ended. The authors also note that LLM-generated trajectories can fail due to non-monotonic jumps, suggesting that the method’s robustness outside supervised-style tasks deserves more discussion and evidence.

**Related work positioning.** The claim that many steps are unnecessary is tightly related to prior work on conditional computation and early exit / dynamic depth in Transformers. The paper would benefit from clearer positioning against these lines and from discussing why DIS provides advantages beyond simply exiting earlier or reducing depth. In addition, multiple recipe changes are bundled together (e.g., fixed supervision steps, removal of halting/ACT, changed recursion budget, timestep conditioning), but ablations that isolate which changes drive performance/efficiency improvements are limited.

---

> ### Author Rebuttal · Authors · 2026-03-30
>
> #### **Answers to Weaknesses**
>
> 1. **Lack of analysis experiments**. ... *the paper could report the distribution of the margin across steps, correlate it with stepwise accuracy/loss improvements, or use it to identify dead-compute steps in practice.*
>
>      We thank the reviewer for this excellent suggestion. To directly demonstrate that the advantage-margin criterion predicts beneficial steps, we analyzed the expected Advantage Margin $\mathbb{E}[A(y^*) - \mathbb{E}_{a\sim\pi_w}A(a)]$ across recursion steps during inference on ARC-1 problem. During successful predictions, this margin increases alongside stepwise accuracy. Conversely, when the margin approaches zero or plateaus, the stepwise accuracy immediately ceases to improve. Following your suggestion we will include this analysis and the corresponding margin distribution plots in the revised paper.
>
> 2. **Generalization to open-domain tasks**. *It's currently unclear how DIS works when targets are ambiguous, multi-solution, or open-ended... method’s robustness outside supervised-style tasks deserves more discussion and evidence.*
>
>     We thank the reviewer for this interesting question. For open-ended or multi-solution tasks, a single static target $y^*$ may be ambiguous. Nevertheless, for such scenarios, extended versions of the task embedding module can be employed. In the current architecture, we utilized the default task embeddings proposed in the TRM paper, which encode the specific task index the model is required to solve. These embeddings are flexible, when multiple potential targets exist for a given input, different identifiers can be applied to construct the appropriate embedding. Our corruption prior is also flexible and can work with various targets as it done in naive diffusion models for image generation.
>
>     Our results regarding LLM-generated trajectories align with our theoretical analysis, the policy-improvement framing directly motivates the necessity of monotonic intermediate targets, which simple semantic generation fails to provide. We agree that further discussion on this topic is necessary, therefore, in the revised paper, we will move the discussions currently in Appendix C into the main part of the paper.
>
> 3. **Related work positioning**. *The claim that many steps are unnecessary is tightly related to prior work on conditional computation and early exit / dynamic depth in Transformers. The paper would benefit from clearer positioning against these lines and from discussing why DIS provides advantages beyond simply exiting earlier or reducing depth.*
>
>     We thank the reviewer for pointing out the relevant papers. While early-exit models (e.g., DeeBERT, PonderNet) improve efficiency by truncating computation when a model is confident, our method does not 'exit early', it supervises every step to act as a strict improvement operator, removing the need for halting classifiers entirely. Following your suggestion, we have included this discussion in the revised manuscript. We would appreciate it if you could suggest specific papers that you feel are most relevant.
>
> 4. *In addition, multiple recipe changes ... ablations that isolate which changes drive performance/efficiency*
>
>      Dear reviewer, regarding the bundled recipe changes, we conducted a strict ablation to isolate the supervision rule. In our paper we didn't add anything, on contrary **we have intentionally decreased the number of total details** and **removed** halting to demonstrate that our results are not the product of specific parameters. Please see our ablation study on 0.8M model for ARC-1 dataset below
>
>     | Model | Halting | Superv. | Recursion | Input Index | Score |
>     | :--- | :---: | :---: | :---: | :---: | :---: |
>     | **DIS** | **No** | **6** | **3** | **Yes** | **24** |
>     | DIS | Yes | 6 | 3 | Yes | 24 |
>     | DIS | No | 6 | 3 | No | 23 (-1) |
>     | DIS | Yes | 16 | 18 | Yes | 26 (+2) |
>     | **TRM** | **Yes** | **16** | **18** | **No** | **12** |
>     | TRM | Yes | 16 | 18 | Yes | 12 |
>     | TRM | Yes | 6 | 3 | Yes | 7 |
>
>     The **bolded lines** in the table represent the default settings reported in the paper.  Our results show that the halting mechanism provides no additional gains when using a small number of supervision steps (6). In contrast, the supervision rule remains highly robust with a score of 24 even without halting, while the TRM model's performance collapses significantly without halting. Increasing the number of latent recursion steps acts as the strongest performance driver for both architectures, even though we reported our results on settings with 18 times fewer forward passes for TRM.
>
> #### **Key Answers For Reviewer:**
>
> 1. See answer 1.
>
> 2. See answer 2
>
> #### **Concluding remarks**
>
> Thank you for your thoughtful feedback and for highlighting areas for improvement. We would appreciate it if you could let us know whether our clarifications address your concerns. We would be happy to discuss them further.

---

> > ### Author Rebuttal · Reviewer_eHog · 2026-04-02
> >
> > I thank the authors for their detailed rebuttal. I'll maintain my positive score of 4.

---

### Official Review · Reviewer_UQtL · 2026-03-14

**Soundness:** 2
**Presentation:** 3
**Significance:** 2
**Originality:** 2
**Overall Recommendation:** 3
**Confidence:** 2

**Summary:**

This paper studies tiny recursive reasoning models and asks what one latent reasoning step is actually doing. The authors argue that a recursive step can be read as a policy-improvement move: the pre-step distribution acts like a reference policy, the post-step distribution acts like an improved policy, and the log-ratio between them behaves like an advantage-style signal. Based on this view, they introduce Deep Improvement Supervision, which replaces supervision only at the final output with intermediate targets produced by a monotone corruption schedule of the ground-truth answer. The experiments focus on N-Queens and ARC-AGI, where the method is claimed to match or outperform TRM while using a much simpler training setup and less compute.

**Compliance With Llm Reviewing Policy:**

Affirmed.

**Key Questions For Authors:**

Question 1: Can the authors compare DIS against simpler forms of intermediate supervision that do not use the policy-improvement interpretation, so the source of the gain is clearer?

Question 2: Have the authors tried intermediate targets that resemble actual reasoning states, instead of a corruption schedule derived from the final answer?

Question 3: Can the paper disentangle the effects of removing halting, shortening recursion, and adding DIS, rather than changing all of them together? i.e., performing the ablation study experiments.

**Limitations:**

Yes

**Strengths And Weaknesses:**

**Strength**

1. This paper attempts to connect latent recursion to policy improvement is interesting, and it is more ambitious than simply proposing another training trick for TRM-style models.

2. DIS itself is easy to understand. Giving the model intermediate targets instead of asking it to discover the whole refinement process from only the final answer is a sensible move, especially in the low-capacity regime the paper cares about.

3. Some of the empirical results are genuinely interesting. The compact ARC result is the clearest example: the jump over compact TRM is large enough that it caught my attention, and the medium setting is also competitive.

4. I also appreciated that the paper is trying to say something about why recursive latent computation may help, rather than only showing one more benchmark table.

**Weakness**

1. The main issue for me is that the interpretation is much stronger than the evidence. The key step in the argument is to treat the post-reasoning distribution as an optimality-conditioned policy, but that is really taken as a modeling assumption. I did not come away convinced that the paper has shown latent recursion in TRM is actually implementing policy improvement in any substantive sense.

2. Relatedly, the DIS story is not well isolated. Once the model is given explicit intermediate targets at every step, it is not surprising that training becomes easier. The paper does not do enough to separate the value of the RL-style interpretation from the value of straightforward deep supervision.

3. I would have liked to see stronger baselines around the supervision choice itself. Right now the paper mostly compares against TRM, but not against simpler stepwise-target alternatives that do not rely on the policy-improvement framing. That makes it hard to know what exactly should be credited for the gains.

4. The choice of intermediate targets also leaves me unconvinced from a reasoning perspective. In the main setup, those targets come from corrupting the final answer on a schedule. That may work as a curriculum, but it is not the same thing as supervising a meaningful reasoning trajectory.

5. The empirical scope is narrow for the kind of claim the paper wants to make. Everything is built around N-Queens and ARC, and even within ARC the picture is mixed: the compact setting looks strong, but the larger comparisons are much less decisive.

6. The presentation needs work. I could follow the main ideas, but the writing is rough, a number of claims are overstated, and the reviewer-facing instructions left in the manuscript are not acceptable in a conference submission.

7. I also think the paper changes too many parts of the original TRM recipe at once. Supervision, halting, recursion budget, and training schedule are all simplified together, which makes attribution harder than it should be.

---

> ### Author Rebuttal · Authors · 2026-03-30
>
> #### **Answers to Weaknesses**
>
> 1. **Interpretation**. *Interpretation is much stronger than the evidence...*
>
>      We want to clarify that our Section 3 formally derives that the log-ratio of the post-reasoning and pre-reasoning distributions mathematically works as an advantage signal. Our main motivation for considering the post-reasoning distribution as optimality-conditioned comes from the fact that reasoning steps are inherently conditioned on the input signal (x). The same mechanism is used to establish the optimality condition in the classifier-free diffusion guidance interpretation using policy improvement. By demonstrating this alignment, we substantiate that the recursion effectively implements a policy improvement operator.
>
> 2. **Paper story**. *Relatedly, the DIS story is not well isolated. Once the model is given explicit intermediate targets at every step, it is not surprising that training becomes easier...*
>
>      Dear reviewer, in our opinion the statement that this is "not surprising" lacks formal grounding, whereas our goal is to demonstrate exactly why this phenomenon occurs. Therefore, to formally explain why it works, our contribution is not the mere addition of an intermediate loss, but the supervision of the *improvement operator itself*. Our method shapes the log-ratio advantage to align with stepwise monotonic targets, operationalizing the RL-style interpretation into a concrete gradient signal.
>
> 3. **Step-wise baselines**.*I would have liked to see stronger baselines around the supervision choice itself... stepwise-target alternatives that do not rely on the policy-improvement framing*
>
>      As detailed in Appendix C, we did explicitly evaluate an alternative stepwise-target mechanism: semantic reasoning trajectories generated by a teacher LLM (Gemini 2.5 Pro). This alternative severely underperformed our discrete diffusion prior.  This met our theoretical analysis, that directly motivates the necessity of *monotonic* intermediate targets, which semantic generation fails to provide. Even if these traces are meaningful and human-aligned, small models struggle to map the highly nonlinear, non-monotonic semantic jumps.
>
> 4. *The choice of intermediate targets also leaves me unconvinced from a reasoning perspective...*
>     We are providing the simplest solution, which can be implemented in a few lines of code, as constructing chains of thought is complicated and costly. Moreover, what a reasoning trajectory and chain of thought should look like in the latent space remains an open question. Please consider the answer above as well (3).
>
> 5. ***The presentation needs work**. *I could follow the main ideas, but the writing is rough, a number of claims are overstated, and the reviewer-facing instructions left in the manuscript are not acceptable in a conference submission.*
>
>      Dear reviewer, **the reviewer-facing instructions were inserted into the final version by the ICML conference organizers and are not an oversight on our part**. We have corrected typos and improved the overall flow. We would appreciate it if you could specify exactly which parts of the paper you found rough or overstated.
>
> 6. **Changes from TRM**. *I also think the paper changes too many parts of the original TRM recipe at once.*
>
>      We did not merely change these parts, **we removed** them entirely, decreasing the recursion budget by 18x. This is a significant distinction. Please see our ablation study on 0.8M model for ARC-1 dataset below
>
>     | Model | Halting | Superv. | Recursion | Input Index | Score |
>     | :--- | :---: | :---: | :---: | :---: | :---: |
>     | **DIS** | **No** | **6** | **3** | **Yes** | **24** |
>     | DIS | Yes | 6 | 3 | Yes | 24 |
>     | DIS | No | 6 | 3 | No | 23 (-1) |
>     | DIS | Yes | 16 | 18 | Yes | 26 (+2) |
>     | **TRM** | **Yes** | **16** | **18** | **No** | **12** |
>     | TRM | Yes | 16 | 18 | Yes | 12 |
>     | TRM | Yes | 6 | 3 | Yes | 7 |
>
>     The **bolded lines** in the table represent the default settings reported in the paper. In our paper we didn't add anything, on contrary we have intentionally decreased the number of total parameters to demonstrate that our results are not the product of specific parameter tuning.
>
>     Our results show that the halting mechanism provides no additional gains with a 6 supervision steps. In contrast, the DIS model remains highly robust with a score of 24 even without halting. The TRM performance collapses under minimal configurations. Increasing the number of latent recursion steps acts as the strongest driver for both architectures. We will extend our ablation for bigger models in revision.
>
> #### **Key Answers For Reviewer:**
>
> 1. Answer 3.
>
> 2. Answer 4.
>
> 3. Answer 6.
>
> #### **Concluding remarks**
>
> Thank you for your thoughtful feedback. We would appreciate it if you could let us know whether our clarifications address your concerns. If there are any remaining issues, we would be happy to discuss them further.

---

> > ### Author Rebuttal · Reviewer_UQtL · 2026-04-05
> >
> > Thank you for the detailed rebuttal. The additional discussion of alternative intermediate supervision and the added ablation table are helpful, and they increase my confidence that DIS captures a real empirical effect rather than only reflecting one fragile training setup. However, my main concern remains unchanged: the policy-improvement interpretation still appears stronger than what the evidence currently supports, and it is still difficult to cleanly attribute the gains to that framing rather than to the benefits of a particular form of intermediate supervision. The new ablation improves the empirical story, but it is limited in scope and does not fully resolve the attribution issue. For these reasons, I am maintaining my overall recommendation at Weak Reject.

---

> > > ### Author Response · Authors · 2026-04-05
> > >
> > > Dear reviewer, we appreciate your engagement with our work and are glad to hear that ablations and discussions have increased your confidence in the empirical effects. We would like to address your remaining concern.
> > >
> > > We do not view the policy-improvement framework as a separate variable claiming independent gain. Rather, we propose it as the explanatory lens that motivated the design of DIS. As discussed in the introduction, current architectures like TRM and HRM demonstrate impressive capabilities but lack a explanation for why latent reasoning steps are effective, or not. The policy-improvement perspective formalizes latent reasoning as an advantage-like signal. This Advantage Margin condition explicitly dictates the need for the structured, monotonically improving targets that DIS provides. In short, DIS is the practical mechanism that drives the gains, but the policy-improvement framework explains why this specific form of supervision effectively reduces forward passes.
> > >
> > > We will revise the abstract, introduction, and conclusion to clearly position the policy-improvement framework as a valuable interpretative lens that inspired our training scheme, rather than an empirically verified absolute. We would appreciate it if you could specify exactly which parts of the paper you found overstated. We hope this commitment to calibrating our claims in the text addresses your core concern, and we thank you again for your help in improving our paper.

---

### Official Review · Reviewer_vCCx · 2026-03-20

**Soundness:** 3
**Presentation:** 3
**Significance:** 3
**Originality:** 3
**Overall Recommendation:** 5
**Confidence:** 3

**Summary:**

The paper examines when and why latent reasoning improve model performance by investigating an important aspect of when iterative reasoning steps meaningfully improve predictions versus becoming wasted computation. They show that each recursion step can be interpreted as transforming a reference policy into an improved one via an advantage-like signal, analogous to reinforcement learning policy updates. Based on this finding, the paper then proposes Deep Improvement Supervision (DIS), a training method that provides stepwise intermediate targets to enforce consistent improvement at each step. Experiments on baselines demonstrate that DIS improves efficiency while achieving comparable performance.

**Compliance With Llm Reviewing Policy:**

Affirmed.

**Final Justification:**

The authors’ rebuttal was helpful and appreciated. It addressed all my major concerns. Overall, it reinforced my view that the paper makes a solid, though somewhat limited, theoretical contribution.

**Key Questions For Authors:**

1. How well does the assumption that the post-reasoning policy approximates an optimality-conditioned policy hold in practice, and in what situations might this approximation break down?
2. How does the proposed method generalize beyond ARC and N-Queen puzzles, particularly to more diverse or real-world reasoning tasks?
3. How sensitive is DIS to the choice of intermediate target generation?

**Limitations:**

yes; If there are any concerns that highlight limitations, the authors may consider including and discussing them.

**Strengths And Weaknesses:**

The paper proposes a novel framework interpreting latent reasoning as a policy improvement process, offering a unifying perspective that connects recursive models with reinforcement learning. It addresses an important inefficiency issue by identifying and explaining “dead compute,” providing a principled understanding of when iterative reasoning steps meaningfully contribute to performance. With extensive experiments, the method outperforms other baselines in efficiency, significantly reducing the number of forward passes while maintaining comparable performance. While these promising results, several concerns could be addressed:

1. The empirical evaluation is somewhat limited, as experiments are mainly conducted on two puzzles, which more complex and large-scale reasoning tasks could also be considered.
2. Comparison fairness is questionable since the proposed method modifies multiple aspects of the baseline, making it unclear whether the observed improvements stem from the new supervision strategy or other architectural and training changes. It is better to include ablation studies on these factors.
3. At the beginning of the paper, the authors state their goal of understanding when reasoning improves performance and when it fails; however, the paper lacks empirical analysis of failure cases and does not discuss scenarios where reasoning leads to negative or improvements.

---

> ### Author Rebuttal · Authors · 2026-03-30
>
> We thank the reviewer for the positive assessment of the novelty, soundness, and practical relevance of our work, and for the constructive suggestions on interpretation and empirical scope. We address each point below.
> #### **Answers to Weaknesses**
>
> 1. **More complex and large-scale reasoning tasks could also be considered.**
>
>      To further broaden the empirical scope and demonstrate robustness, we have conducted and included new experiments on the Sudoku dataset in the revised paper, (will show them during discussion period) we understand that this is still an another puzzle dataset, however this dataset is considering as strong baseline, importantly on which LLMs are performing poorly.
>
> 2. **Comparison fairness is questionable since the proposed method modifies multiple aspects of the baseline..**
>
>      Dear reviewer, please see our ablation study on 0.8M model for ARC-1 dataset below
>     | Model | Halting | Superv. | Recursion | Input Index | Score |
>     | :--- | :---: | :---: | :---: | :---: | :---: |
>     | **DIS** | **No** | **6** | **3** | **Yes** | **24** |
>     | DIS | Yes | 6 | 3 | Yes | 24 |
>     | DIS | No | 6 | 3 | No | 23 (-1) |
>     | DIS | Yes | 16 | 18 | Yes | 26 (+2) |
>     | **TRM** | **Yes** | **16** | **18** | **No** | **12** |
>     | TRM | Yes | 16 | 18 | Yes | 12 |
>     | TRM | Yes | 6 | 3 | Yes | 7 |
>
>     The **bolded lines** in the table represent the default settings reported in the paper. In our paper we didn't add anything, on contrary we have intentionally decreased the number of total parameters to demonstrate that our results are not the product of specific parameter tuning.
>
>     Our results show that the halting mechanism provides no additional gains when using a small number of supervision steps (6). In contrast, the DIS model remains highly robust with a score of 24 even without halting, while the TRM model's performance collapses significantly under minimal configurations. Increasing the number of latent recursion steps acts as the strongest performance driver for both architectures, even though we reported our results on settings with 18 times fewer forward passes for TRM.
>
> 3. **The paper lacks empirical analysis of failure cases and does not discuss scenarios where reasoning leads to negative or improvements.**
>
>      Our interpretation is mostly theoretical, our main claim defines a failure case (dead compute) as a step where the log-ratio advantage margin fails to align with the target. To empirically ground this, following your comment, the revised paper now includes an analysis tracking the expected Advantage Margin $\mathbb{E}[A(y^*) - \mathbb{E}_{a\sim\pi_w}A(a)]$ during inference. When this margin approaches zero, obviously stepwise accuracy plateau or degrades.
>
> #### **Key Answers For Reviewer:**
>
> 1. *How well does the assumption that the post-reasoning policy approximates an optimality-conditioned policy hold in practice, and in what situations might this approximation break down?*
>
>      Our Proposition 4.1 formally guarantees that minimizing the DIS loss continuously drives the Advantage Margin positive, effectively enforcing this optimality condition. So the approximation holds robustly when the target trajectory is monotonic. The approximation breaks down exclusively when intermediate targets contain non-monotonic jumps. In appendix C, we discuss the experiment when models are trained on highly nonlinear semantic trajectories (e.g.,solution trajectories generated by LLM. We found that the model cannot map the intermediate states, as the policy improvement guarantees are nullified. Please see more discussions on this in Q3.
>
> 2. See Answer 1.
>
>
>
> 3. *How sensitive is DIS to the choice of intermediate target generation?*
>
>     Our theoretical derivation mandates a strictly improving target sequence. To test this sensitivity empirically, we replaced the discrete diffusion prior with LLM-generated semantic reasoning traces. For each training sample in the ARC-1 task, we ran Gemini 2.5-Pro to generate a reasoning trajectory between the given input and target. The model's performance degraded sharply from 24% to 10% accuracy. Small, parameter-efficient latent models struggle to process the highly nonlinear "jumps" inherent in semantic generation. This confirms our theoretical claim that, for effective improvement, mathematically smooth and monotonic target schedules are necessary. We will move this discussion from Appendix C to the main part of the paper.

---

> > ### Author Rebuttal · Reviewer_vCCx · 2026-04-04
> >
> > The authors have addressed most of the concerns. I have no major remaining concerns.

---

### Decision · Program_Chairs · 2026-04-30

**Decision:**

Accept (regular)

**Comment:**

This paper studies tiny reasoning models with latent recursion. The main technical contribution is that latent reasoning can be viewed as a classifier-free guidance and policy improvement algorithm. Based on this insight, the authors propose an RL algorithm for training these models. The approach is evaluated empirically on ARC benchmarks. The authors report that tiny models with a few million parameters can match the performance of models with billions of parameters.

This paper makes a solid technical contribution that has a potential to be followed. The reviewers gave the paper scores 2x Weak Reject, Weak Accept, and Accept. The rebuttal addressed two main concerns:

* **Overselling:** This paper focuses on *reasoning in tiny recursive models (TRMs)*. The authors need to make it clear, including in the title. My suggested title is *Latent Reasoning in TRMs is Secretly a Policy Improvement Operator*. Without the title change, potential readers will think that this paper is about math reasoning, which is not the case. This threw reviewers off balance. This will also reduce the scope of the claims, which is needed to be more honest about the contribution.

* **Experiments:** The authors conducted new experiments on Sudoku dataset and will add them to the paper. While this is another puzzle dataset, it goes beyond ARC. In addition, the authors reported the performance of their method and TRM for a range of hyper-parameters, showing that the reported improvements are robust.

This is a good paper that brings a new perspective. The authors should incorporate my and reviewer's comments to avoid overselling and be more honest about their contribution.